# Palmitoylethanolamide (PEA) Inhibits SARS-CoV-2 Entry by Interacting with S Protein and ACE-2 Receptor

**DOI:** 10.3390/v14051080

**Published:** 2022-05-17

**Authors:** Rossella Fonnesu, Venkata Bala Sai Chaitanya Thunuguntla, Ganesh Kumar Veeramachaneni, Jayakumar Singh Bondili, Veronica La Rocca, Carolina Filipponi, Pietro Giorgio Spezia, Maria Sidoti, Erika Plicanti, Paola Quaranta, Giulia Freer, Mauro Pistello, Michael Lee Mathai, Michele Lai

**Affiliations:** 1Retrovirus Center, Department of Translational Research and New Technologies in Medicine and Surgery, University of Pisa, 56100 Pisa, Italy; fonnesurossella@gmail.com (R.F.); veronica.larocca@santannapisa.it (V.L.R.); carolina.filipponi@gmail.com (C.F.); piergiorgiospezia@gmail.com (P.G.S.); maria.sidoti.996@gmail.com (M.S.); e.plicanti@studenti.unipi.it (E.P.); paola.quaranta@unipi.it (P.Q.); giulia.freer@unipi.it (G.F.); mauro.pistello@unipi.it (M.P.); 2Institute of Health and Sport, Victoria University, Melbourne, VIC 8001, Australia; venkata.thunuguntla@live.vu.edu.au (V.B.S.C.T.); michael.mathai@vu.edu.au (M.L.M.); 3Department of Biotechnology, Koneru Lakshmaiah Education Foundation, Vaddeswaram 522502, India; ganesh.vgk55@gmail.com (G.K.V.); jksingh@kluniversity.in (J.S.B.)

**Keywords:** SARS-CoV-2, bioactive lipids, nutraceutical, antiviral, spike protein, PEA, palmitoylethanolamide

## Abstract

Lipids play a crucial role in the entry and egress of viruses, regardless of whether they are naked or enveloped. Recent evidence shows that lipid involvement in viral infection goes much further. During replication, many viruses rearrange internal lipid membranes to create niches where they replicate and assemble. Because of the close connection between lipids and inflammation, the derangement of lipid metabolism also results in the production of inflammatory stimuli. Due to its pivotal function in the viral life cycle, lipid metabolism has become an area of intense research to understand how viruses seize lipids and to design antiviral drugs targeting lipid pathways. Palmitoylethanolamide (PEA) is a lipid-derived peroxisome proliferator-activated receptor-α (PPAR-α) agonist that also counteracts SARS-CoV-2 entry and its replication. Our work highlights for the first time the antiviral potency of PEA against SARS-CoV-2, exerting its activity by two different mechanisms. First, its binding to the SARS-CoV-2 S protein causes a drop in viral infection of ~70%. We show that this activity is specific for SARS-CoV-2, as it does not prevent infection by VSV or HSV-2, other enveloped viruses that use different glycoproteins and entry receptors to mediate their entry. Second, we show that in infected Huh-7 cells, treatment with PEA dismantles lipid droplets, preventing the usage of these vesicular bodies by SARS-CoV-2 as a source of energy and protection against innate cellular defenses. This is not surprising since PEA activates PPAR-α, a transcription factor that, once activated, generates a cascade of events that leads to the disruption of fatty acid droplets, thereby bringing about lipid droplet degradation through β-oxidation. In conclusion, the present work demonstrates a novel mechanism of action for PEA as a direct and indirect antiviral agent against SARS-CoV-2. This evidence reinforces the notion that treatment with this compound might significantly impact the course of COVID-19. Indeed, considering that the protective effects of PEA in COVID-19 are the current objectives of two clinical trials (NCT04619706 and NCT04568876) and given the relative lack of toxicity of PEA in humans, further preclinical and clinical tests will be needed to fully consider PEA as a promising adjuvant therapy in the current COVID-19 pandemic or against emerging RNA viruses that share the same route of replication as coronaviruses.

## 1. Introduction

SARS-CoV-2 infection causes the induction of an uncontrolled release of cytokines and other pro-inflammatory mediators that are found in severely ill COVID-19 patients [1,2,3]. Indeed, the virus causes an abnormal activation of macrophages that are responsible for acute respiratory distress syndrome and the subsequent deaths of COVID-19 patients [4]. This is characterized by an increased infiltration of committed macrophages and their Th2/Th17 programming, leading to mortality. Once derailed, hyperactive macrophages secrete high levels of IFN-γ, IP-10, IL-6, IL-17, and TNF-α, along with TGF-β and other interleukins, which contribute to the severity of COVID-19 and lung injury [5].

Many immunosuppressive drugs have been proposed to counteract the pathogenicity of SARS-CoV-2, with encouraging results [6,7,8]. Nonetheless, dampening the inflammatory response during infection may impair the optimal host response and predispose patients to secondary infections. In keeping with this, a recent single-center retrospective case review analyzed the clinical outcomes and secondary infection rates in COVID-19 patients receiving immunosuppressive treatments and described a significantly higher mortality rate and occurrence of secondary infections, particularly by *S. pneumoniae* [9,10]. Diet and nutrition are receiving growing interest by the public, given the compelling evidence of their pivotal role in modulating immune function [11,12]. Indeed, functional foods may bear the potential to increase host defenses against viral infections [13,14]. In this context, many food supplements have been proposed by the scientific community and the general public, yielding conflicting information on this topic [15,16]. The ideal nutraceutical should have proven immunomodulatory activities, and, at the same time, it should be tested for its pharmacokinetic and pharmacodynamic properties and for its efficacy against infections in clinical settings. Since SARS-CoV-2 was identified, researchers immediately began searching for targets to inhibit viral infectivity or replication by modelling molecular interactions that would inhibit the entry of the virus into cells or the viral proteins necessary for enzymatic functions and structural assembly. Libraries of new and existing drugs [17,18] were screened to find candidate molecules that would inhibit these viral functions. One of the candidate molecules was palmitoylethanolamide (PEA), one of the members of the N-acyl-ethanolamine family. PEA was identified more than five decades ago and was shown to reduce allergic reactions and inflammation in animals in addition to influenza symptoms in humans. Interest in this compound faded, however, until the discovery that one of its structural analogs, anandamide (arachidonoylethanolamide), serves as an endogenous ligand for cannabinoid receptors, the molecular target of Δ9-tetrahydrocannabinol [19]. Since this finding, PEA has been shown to inhibit peripheral inflammation and mast cell degranulation as well as to exert neuroprotective and antinociceptive effects in rats and mice [20]. Recently, it was discovered that the anti-inflammatory activity of PEA does not follow the same route as anandamide. Instead, PEA-induced analgesic and anti-inflammatory activities are mediated by the activation of peroxisome proliferator receptor alpha (PPAR-α) [21,22]. The efficacy of PEA in the prevention or treatment of bacterial and viral infections has also been reported [21]. This encouraging evidence in the literature has stimulated research as to whether PEA can be used to inhibit the pathogenesis of SARS-CoV-2 [23,24].

In this study, we used in silico testing to probe the ability of PEA to inhibit the binding of the SARS-CoV-2 spike (S) protein to its target receptor, the ACE-2 enzyme, using methods previously described [18]. Then, we tested the ability of PEA to inhibit viral infection and replication in several cellular models to further shed light on its mechanism of action. The study aims to verify the potential antiviral effect of PEA against SARS-CoV-2 and its emerging variants of concern (VOC), such as Delta or Omicron. The antiviral role of PEA was tested on both viral entry and replication. Indeed, PEA might exert a dual antiviral activity by binding to the SARS-CoV-2 S protein and activating PPAR-α signaling, which, once active, dismantles intracellular lipid droplets (LDs), the very same vesicles that have recently been suggested to boost SARS-CoV-2 replication [25].

## 2. Materials and Methods

### 2.1. In Silico Methodology

#### 2.1.1. Protein Preparation

The PDB (Protein Data Bank) structure of the novel coronavirus (PDB ID: 6LZG) was retrieved and further processed using the protein preparation wizard. The parameters used in refining the structure were the addition of hydrogens, creating disulfide bonds, maintaining zero-order bonds, and converting seleno-methionine to methionine in the import and process tab. Further processing in the refine tab optimized the hydrogen bonds to repair and finally minimize the structure through force field OPLS_2005 [26].

#### 2.1.2. Molecular Docking Studies

Initially, the molecular models of PEA were prepared using the LigPrep module. Further, a grid was generated by fixing an active site with the residues Gln 493, Gln 498, Asn 487, Tyr 505, and Lys 417 present in the spike protein, which were designated as the crucial residues in binding with the ACE2 receptor. Using the receptor grid, the molecules were docked following the Glide XP docking protocol [27,28,29].

#### 2.1.3. Molecular Dynamic Simulations (MDS)

Molecular dynamic simulations (MDS) of the complexes were performed using Desmond software. Initially, the complex was imported into the system builder application of the Desmond module using default parameters, such as the SPC (simple point-charge) solvent model, orthorhombic periodic boundary box, and minimizing the volume, and a model system was generated for simulations. Continuing with the ions tab of the system builder application, Na+ ions were added based on the total charge, and a salt concentration of 0.15 M was also added to neutralize the system. The second step in the simulation protocol was minimization; the complex obtained from the system builder was relaxed by setting the maximum iterations number to 2000, and the remaining parameters were set to default. Finally, the minimized complex was subjected to molecular dynamic simulations by setting the ensemble parameter to NPT (isothermal–isobaric ensemble, number of particles (N), Pressure (P), and Temperature (T)), 300 K temperature, and 1 bar pressure. The simulation run time was set to 200 ns [30].

#### 2.1.4. Cell Culture and Treatments

Huh-7, Vero E6, Calu-3, and HEK-293T cells were purchased from American Type Culture Collection (Manassas, USA) and cultured at 37 °C and 5% CO_2_ in Dulbecco’s Modified Eagle’s Medium (DMEM) supplemented with 10%, 5%, or 2% fetal bovine serum (FBS), 2 mM L-glutamine, and antibiotics (penicillin and streptomycin). Micronized palmitoylethanolamide, a formulated highly absorbable form of PEA and Levagen+, incorporating the LipiSpperse delivery system (mPEA, Levagen+, and Gencor pacific) was used from 100 μM to 0.25 μM to treat Huh-7 cells 30 min before and after viral infection by incubation for 30 min with SARS-CoV-2 before the inoculum and using a combination of those conditions above. The inocula were removed by three washings with PBS. The PPAR-α antagonist GW6471 (Sigma-Aldrich, St. Louis, MO, USA) was administered to cells 30 min prior PEA treatment. All cell cultures were tested for mycoplasma contamination [31]. Cell viability assays were performed as follows: cells were plated (2 × 10^4^ cells/well) in 96-well plates and incubated overnight. Then, cells were incubated with PEA at various concentrations for 24 h, and their viability was assessed using a WST-8 (Sigma-Aldrich) assay according to the manufacturer’s instruction.

#### 2.1.5. Infections

The infection of Huh-7 cells was performed with 0.1 MOI of clinical strains of SARS-CoV-2 VR PV10734, B.1.617.2, and B1.1.529 obtained from the U.O. of Virology, AOUP, Pisa, Italy. The infection of Huh-7 cells was performed with 0.1 MOI of VSV (Mudd–Summers isolate) obtained from D. Kolakovsky, University of Geneva (Geneva, Switzerland), following the exact condition of SARS-CoV-2 assays. The infection of Huh-7 cells with HSV-2 was performed with 0.1 MOI of HSV-2 obtained from the U.O. of Virology, AOUP, Pisa, Italy. All experiments using SARS-CoV-2 were performed under biosafety level 3 protocols, following containment procedures approved at the Laboratory of Virology Unit, Pisa University Hospital.

#### 2.1.6. Lentiviral Vectors

HEK-293T cells were used to produce GFP-bearing lentiviral vectors pseudotyped with SARS-CoV-2 S protein (Spike-PLVs). The cells were transduced with four different plasmids, pNL-EGFP, pSPAX2, pCMV-Rev, and pCMV14-3xFlag-SARS-CoV-2 S (#17579, #12260, #119322, and #145780, Addgene, Watertown, MA, USA). The lentiviral vectors were harvested 72 h after transfection. Spike-PLVs were used to test the capacity of PEA to bind to S protein and ACE-2 receptors. Spike-PLVs were incubated with PEA 1, 10, and 100 μM right before Huh-7 transduction. After, Huh-7 cells were analyzed with confocal microscopy 72 h post transduction, and positive cells were identified using high-content confocal screening with the following building blocks: find nuclei, find cytoplasm, calculate intensity properties (GFP), and select population (GFP+).

#### 2.1.7. RNA Extraction and Real-Time PCR

Total RNA from the cellular substrate was extracted using QIAzol Lysis Reagent according to the manufacturer’s instructions (QIAGEN, Hilden, Germany). Briefly, 200 ng of RNA were amplified using a One Step PrimeScriptTM III RT-qPCR Mix kit (Takara Bio, Kyoto, Japan) using the following primers: SARS-CoV-2 RdRp forward 5′-TCACCTATTTTAGCATGGCCTCT-3’, reverse 5′-CGTAGTGCAACAGGACTAAGC-3′, probe 5′-/56-FAM/TGCTTGTGCCCATGCTGC-3′; β-actin forward 5′-AAGGAGAAGCTGTGCTACGTC-3’, reverse 5′-AGACAGCACTGTGTTGGCGTA-3’, probe 5’-/56-FAM/TGGCCACGGCTGCTTCCA. VSV genomes were amplified by real-time PCR using a QuantiNova SYBR Green RT-PCR kit (QIAGEN^®^, Hilden, Germany) using the following primers: forward: 5′- TTCAATGAAGATGACTATGCCACAAGAG3′, reverse: 5′AAGAACTCCATCCCAGTTCTTACTATCC3′. Viral DNA of HSV-2 was extracted from supernatants 24 h post-infection using QIAamp DNA Mini and Blood Mini kits (QIAGEN^®^, Hilden, Germany). Then, the HSV-2 genome was amplified by real-time PCR by using a QuantiNova SYBR Green PCR kit (QIAGEN^®^, Hilden, Germany) using the following primers: forward: 5′-ATCAACTTCGACTGGCCCTT-3′; reverse 5′ CCGTACATGTCGATGTTCAC-3′. ACE-2: forward: 5′-TCCATTGGTCTTCTGTCACCCG-3′, reverse: 5′-AGACCA TCCACCTCCACTTCTC-3′. The relative quantity of gene expression was calculated by the 2-ΔΔCt method using β-actin expression as a housekeeping gene.

#### 2.1.8. Virus Yield Reduction Assay

Huh-7 cells were seeded in a 12-well plate and grown to confluence. Viral suspensions were incubated or not for 30 min at various concentrations of PEA (100, 10, 5, 1, and 0.25 μM) then were used to infect cells. Infection was carried out for 1 h. The cell monolayer was covered with a fresh medium of 2% FBS. Cell supernatants were taken after 24 h. Then, the cell supernatants obtained from each sample were seeded in duplicate in a 96-well plate of Vero E6 (2 × 10^4^) plated the day before. Every viral inoculum was serially diluted 1:3 eight times down the plate. After 1.5 h of infection, the viral inoculum was discarded, and fresh medium containing 2% carboxymethylcellulose (CMC, Sigma-Aldrich) was added. Cells were fixed with 4% paraformaldehyde after 24 h. CMC was then removed, and the monolayer was stained with crystal violet. The viral endpoint dilution was determined for each sample, as described elsewhere.

#### 2.1.9. High-Content Confocal Imaging

Imaging experiments were performed using an Operetta CLS high-content imaging device (PerkinElmer, Hamburg, Germany) and were analyzed with Harmony 4.6 software (PerkinElmer). Huh-7 cells were seeded in 96-well CellCarrierUltra plates (Perkin Elmer, Hamburg, Germany) 24 h prior to treatment or infection, as described above. Cells were fixed in ice-cold methanol and stained 48 h after infection with the following primary antibodies: Anti-SARS-CoV-2 Spike protein (#40588 RC02, Sino Biological, Beijing, China, 1:200). Lipid droplets (LDs) were stained using Oil Red O (1:5000 diluted in water) for 15 min (Sigma-Aldrich, St. Louis, MO, USA). Nuclei were stained with DAPI (1 µg/mL). To quantify the number of infected cells, the following building block was used: find nuclei, find cytoplasm, calculate intensity properties (SARSCoV2-S1-Alexa 488), select population of infected cells, and find SARS-CoV-2-positive (Alexa 488+) cells. Around 140 fields per well per experiment were analyzed using a 63× water objective. To quantify the number of LDs, we used a detailed protocol described previously [32,33].

#### 2.1.10. Western Blot Analysis

Huh-7 cells were lysed with RIPA lysis buffer (Millipore, Burlington, MA, USA). Membranes were incubated at 4 °C overnight with the following antibodies: anti-SARS-CoV-2 N (1:1000, MA14AP1502, Sino Biological, Beijing, China) and anti-β-actin (1:1000, A2066 Sigma-Aldrich, St. Louis, MO, USA). Blots were acquired and analyzed using a Chemidoc XRS system (BioRad, Berkeley, CA, USA).

#### 2.1.11. Cell-Based ELISA Assay

Hek 293T and Hek 293T ACE-2 cells were seeded in 96-well plates coated with poly-d-lysine (Sigma-Aldrich) at a density of 4 × 10^4^ cells. Then, the cells were treated with SARS-CoV-2 spike RBD-mFc recombinant protein (Sino Biological) at 100 ng/mL for 30 min at 4 °C in the presence or absence of an equimolar dose of PEA. The cells were washed with PBS and fixed in 4% paraformaldehyde for 20′ at 4 °C. Subsequently, the cells were washed in PBS and treated with 0.1 M glycine and 3% hydrogen peroxide. Then, the cells were incubated in a blocking solution (PBS containing 5% FBS and 1% BSA) for 30 min at room temperature. Thereafter, cells were labeled for 1 h with a horseradish peroxidase (HRP)-conjugated antibody against the mFc tag of the RBD-mFc recombinant protein (1:20,000) diluted in blocking solution. After four washes in PBS, the HRP substrate containing 3,30,5,50-tetramethylbenzidine substrate was added to each well and incubated for 15–30 min until color development. To stop the reaction, 1M HCl was added, and the absorbance was read at 450 nm. Cell content normalization was performed by staining cells with DAPI and reading the corresponding fluorescence.

#### 2.1.12. Data Analysis

All graphing and statistical tests were performed in Prism Graphpad (version 8, https://www.graphpad.com/). Data were expressed as means ± SD (* *p* < 0.05, ** *p* < 0.01, ***, *p* > 0.001, *p* < 0.0001). All results were obtained from at least three independent experiments and were expressed as means ± standard errors of the mean (SEM).

## 3. Results

### 3.1. Molecular Docking Studies of ACE2 Receptor with PEA

In the PDB structure, chain A represents the ACE 2 receptor and chain B denotes the spike protein. The present study was intended to validate the PEA molecular interaction with the spike protein and hence chain B was considered for the studies, omitting chain A. Previous studies [18] revealed that Gln 493, Gln 498, Asn 487, Tyr 505, and Lys 417 residues in the receptor-binding domain (RBD) site of the spike protein were crucial in binding with the ACE2 receptor in forming the ACE2–spike complex. By selecting these residues in the receptor grid generation protocol, the grid was generated and further docked with the PEA molecule. The docking studies illustrated in Figure 1 revealed the interaction profile between the spike protein RBD site residues and the PEA molecule. Three hydrogen bonds were identified in the spike–PEA complex.

Our results have shown that PEA acts in a number of ways to inhibit viral infection and replication. Thus, while vaccines remain the frontline method of reducing the impact of the virus on health, their efficacy has been altered by the changes in the spike protein conformation that have come with each variant as they progressed from Alpha to Gamma, Delta, Mu, and Lambda. Indeed, the emergence of the Omicron variant has raised concerns that the large number of conformational changes in the spike protein will render the first series of vaccines less useful. Reassuringly, the vaccines have been joined by a growing number of antiviral treatments that target the function of viral proteins. We have shown in this study that PEA possesses the ability to inhibit both infection and replication aspects of the virus. This means that it has a high probability of retaining antiviral efficacy on the different variants [34] that evolve from Alpha to Omicron and beyond.

The hydrogen bond interaction profiles showcased by the GLN 409 residue with =O and the other two hydrogen bonds were shared between the ARG 408 and ASP 405 residues with the OH group of the PEA molecule. After docking, the molecule was having no interactions with the crucial resides mentioned in the RBD site, but the molecule was found to be covering some of the crucial resides. Earlier studies from Suresh Kumar et al. in 2021 observed that there are no RBD mutations found in the 405, 408, and 409 residues, indicating the efficacy of PEA binding to the RBD domain remains unchanged with the delta (2 mutations) and Omicron (15 mutations) variants [35]. Based on the docking results, it was predicted that the molecule would cover the RBD site residues further, thereby inhibiting binding with the ACE2 receptor. To validate whether the molecule was fixing in the same position as depicted in the docking studies, molecular dynamics simulations were performed. The difference between the docking and simulation studies was that in the docking studies the protein receptor was fixed, i.e., rigid, but the ligand molecule was flexible, whereas in the simulations both the ligand and protein were flexible. The spike RBD–PEA complex was simulated using the parameters as mentioned for a period of 200 ns. The RMSD (root mean square deviation) and RMSF (root mean square fluctuation) were calculated along with ligand interaction profile. After the simulation studies, the molecule lost all the interactions made during the docking studies. This denotes the molecule was relaxed and flexible enough to make orientations inside the active pocket. Initially, the interactions made during the docking studies were sustained for a few nanoseconds, and based on the simulation run, the interactions disappeared because of the PEA orientation. Alternatively, the PEA molecule made interactions with other residues, such as GLY 404, GLU 406, ARG 408, GLY 504, etc., indicating that the molecule was present within the active site, as shown in Figure 2.

The ligand RMSD varied between 4 and 38 Å. In the initial simulation run time, the molecule showed the highest deviations around 38 Å, and later on the deviations were stabilized between 12 and 16 Å from 50 ns on. The ligand deviations were very high during the simulation run, indicating that the molecule became more relaxed and generated several orientations. Similarly, the protein deviations were between 1.2 and 3.2 Å, and the majority were between 2.0 and 2.4 Å, as observed from 100 ns on, which are in the acceptable range. Based on the deviations, the protein was more stable compared to the ligand molecule. The residues in the spike RBD site fluctuated between 0.6 and 4.2 Å. The majority of the residues fluctuated below 1.5 Å, which denotes that the stability of the protein was greater during the simulations.

### 3.2. PEA Decreases SARS-CoV-2 RBD Binding to ACE2 Receptor

With the aim to confirm the predicted binding of PEA with the SARS-CoV-2 S protein localized in the RBD region, we took advantage of a cell-based ELISA assay that is schematically illustrated in Figure 3a. Briefly, we exposed 293T-ACE2 cells to SARS-CoV-2 recombinant RBD protein that was fused with murine Fc with or without equimolar concentrations of PEA. Then, the cells were washed, fixed, and stained with labeled α-mouse antibody. As shown in Figure 3b, PEA decreases RBD binding with ACE2 by ~50% compared to the mock-treated counterparts. As expected, no signal was detected in 293T WT cells.

### 3.3. PEA Reduces S-Pseudotyped Lentiviral Vector Transduction Efficiency

First, we tested PEA toxicity in both Huh-7 and 293T cells, as shown in Figure 4a. No toxicity was detected, even when PEA was present as high as 100 μM in both cell lines. Then, we confirmed the efficient infection of Huh-7 with SARS-CoV-2, which was known before [36]. As shown in Figure 4b,c, these cells can be efficiently infected by SARS-CoV-2 and express high levels of ACE-2 mRNA before and after viral infection.

To probe whether PEA might decrease SARS-CoV-2 viral entry, we generated GFP-bearing lentiviral vectors pseudotyped with SARS-CoV-2 S protein. These vectors were incubated with PEA at 1, 10, and 100 μM, then were used to transduce Huh-7 cells. Positively transduced cells expressing GFP protein were enumerated 72 h post-transduction by high-content confocal microscopy screening. As shown in Figure 4, the administration of PEA at 1 and 10 μM reduced the number of transduced cells by 37% (*p* < 0.001) and 17.9% (*p* < 0.01), respectively. Interestingly, PEA administered at 100 μM did not reduce transduction (Figure 4d,e).

### 3.4. PEA Decreases SARS-CoV-2 Infectivity

Prompted by the previous results on pseudotyped lentiviral vectors, we tested PEA against SARS-CoV-2. With the aim to better elucidate the different interactions proposed by in silico studies, we tested three different PEA treatments, as schematically represented in Figure 5a. First, to probe PEA’s indirect antiviral activity, we treated Huh-7 cells with PEA for 30′ prior to infection. Then, the number of viral genomes were counted at 72 h post-infection by qRT-PCR. Pre-treatment of Huh-7 with PEA at 1 μM and 10 μM reduced the number of SARS-CoV-2 virions released by 39.8% and 40.6%, respectively, compared to untreated cells (Figure 5b). Then, we assessed whether PEA might bind to SARS-CoV-2 S protein RBD and decrease its entry into Huh-7 cells. As predicted, mixing PEA with the virus prior to infection significantly decreased SARS-CoV-2 infectivity. Indeed, as shown in Figure 5c, center panel, the SARS-CoV-2 genomic content in cellular lysates abruptly dropped by 70% when PEA was administered in the range of 0.25–10 μM compared to untreated cells. We also observed that the administration of PEA at 100 μM did not exert any significant reduction. Finally, we pretreated Huh-7 cells with PEA, administered at 1 or 10 μM, before the infection. Then, we infected the cells with SARS-CoV-2 virions incubated or not with PEA at 1 μM. The combined treatment reduced the amount of SARS-CoV-2 genomes by ~64%, as detected by qRT-PCR 72 h after infection (Figure 5d, right panel). Finally, as shown in Figure 5e, PEA was more efficient at reducing SARS-CoV-2 infection when delivered with the virus than when administered to cells prior to infection.

To confirm these results, we repeated the same experimental conditions using high-content confocal microscopy screening. Briefly, cells were seeded in 96-well plates, then treated or not with PEA at 1 or 10 μM and then infected with SARS-CoV-2, incubated or not with PEA at 1 μM. As shown in Figure 6a,b, pre-treatment with PEA at 10 and 1 μM reduced the number of infected cells by 48.8% (*p* < 0.01) and 78.5% (*p* < 0.001), respectively, compared to untreated cells. Interestingly, the combined treatment of PEA administered on both Huh-7 cells and SARS-CoV-2 reduced the number of infected cells by 55.02% (*p* < 0.0001) when PEA was administered at 1 μM, while PEA at 10 μM reduced the amount of infected cells by 89.8% (*p* < 0.0001) compared to controls (Figure 6c).

To further confirm the antiviral activity of PEA, we decided to perform the same analysis shown in Figure 6 on Calu-3 cells, another cell line that is widely used for SARS-CoV-2 infections [37]. As shown in Figure 7a, we performed high-content confocal microscopy screening on Calu-3 cells infected with SARS-CoV-2 mixed or not with PEA (1 μM). To better elucidate the mechanism of action of PEA as an indirect antiviral agent, we also added a potent antagonist of the PPAR-α receptor (GW6471) as a control. As shown in Figure 7a,b, PEA decreased the number of infected cells together with the intensity of expression of SARS-CoV-2 S protein. On the contrary, the administration of GW6471 increased the amount of SARS-CoV-2 S protein, detected as relative fluorescence units (RFU), compared to the PEA-treated or mock counterparts.

### 3.5. PEA Exerts Its Antiviral Activity on SARS-CoV-2 Emerging Variants

The SARS-CoV-2 Delta and Omicron variants have sequentially replaced the Wuhan strain worldwide, becoming the prevalent VOCs. Both variants accumulated several mutations on the S protein that could affect the binding with PEA described above. For this reason, we tested PEA against these two variants. Figure 8a shows that PEA-pretreated Huh-7 cells at 10 and 1 μM reduced the numbers of viral genomes by 62.4% (*p* < 0.0001) and 51.2% (*p* < 0.001) for SARS-CoV-2 Delta and by 43.4% and 77.3% for the Omicron variant, respectively. Interestingly, we show that SARS-CoV-2 Delta and Omicron virions exposed to PEA showed a reduction in viral genomes/cell up to nearly 65% for both VOCs with 10 μM PEA, as quantified by qRT-PCR. (Figure 8b). We also probed the combined administration of PEA to both cells and virions. PEA administered at 1 μM led to significant reductions in the number of viral genomes in cell lysate that were quantified as 75.3% (*p* < 0.0001) and 72.5% (*p* < 0.01) for the Delta and Omicron VOCs, respectively. We noticed that PEA administered at 10 μM was less effective against the Omicron VOC, leading to a 57.3% reduction in viral titer (*p* < 0.01), compared to 71.9% (*p* < 0.0001) for Delta. (Figure 8c).

### 3.6. PEA Does Not Block VSV and HSV-2 Entry in Huh-7 Cells

To probe the specificity of PEA antiviral activity, we exposed vesicular stomatitis virus (VSV) and herpes simplex virus-2 (HSV-2) to PEA at 1 and 10 μM for 30′. Both viruses use different glycoproteins to mediate viral entry, recognizing different cellular receptors compared to SARS-CoV-2. Moreover, the replication of both viruses follows two distinctive paths that differ from the one used by SARS-CoV-2. As shown in Figure 9a–d, PEA did not affect viral entry for either virus, measured as plaque forming units/mL or the qRT-PCR quantification of viral genomes in the supernatants. These results corroborate the hypothesis that PEA binding is specific for SARS-CoV-2 S protein.

### 3.7. PEA Decreases the Amount of Intracellular Lipid Droplets

We hypothesized that the antiviral activity of PEA administered to cells might also depend on the activation of PPAR-α. Indeed, PPAR-α generates a signaling cascade that leads to the disruption of lipid droplets by the activation of β-oxidation within mitochondria and peroxisomes, and the concomitant stimulation of omega-oxidation in microsomes [38]. LDs, indeed, are required by SARS-CoV-2 to fuel its replication. To probe our hypothesis, PEA was administered to Huh-7 cells prior to infection. Then, cells were fixed and stained with Oil Red O to mark LDs. The quantification of the LD content/cell by high-content confocal microscopy revealed that the administration of PEA at 5 μM drove a reduction in LD numbers by 40.3% (*p* < 0.001) in SARS-CoV-2 cells compared to their untreated counterparts (Figure 10a–c).

## 4. Discussion

The spread of SARS-CoV-2 worldwide represents a threat to global public health. In this context, we face the absence of specific antiviral drugs and the urgency of novel and effective treatments against SARS-CoV-2 and future coronaviruses sharing the same entry mechanism. More generally, given the hastiness with which drugs are required during sudden pandemics, it would be highly desirable to repurpose drugs already used for clinical treatments that do not require further toxicity studies. Endocannabinoid-related compounds are endogenous bioactive lipid amides with pleiotropic homeostatic properties, including immune response regulation, control of food intake, neuroprotection, and inhibition of pain and inflammation [39,40,41,42,43,44,45]. These well-known multifaceted properties, readily translatable to clinics, and the lack of unwanted side effects have already attracted the attention of the scientific community toward the repurposing of these compounds during the COVID-19 pandemic [43,46,47,48]. In particular, Oleoylethanolamide (OEA), cannabidiol, PEA, and other unsaturated fatty acids have been selected as drug candidates for potential novel strategies against COVID-19 [47,49,50]. All these compounds share similar characteristics, being endogenous lipids taking part in the host’s immune response to a variety of stimuli that include viral infections [51]. Indeed, PEA has been used in several placebo-controlled double-blind clinical trials on influenza and the common cold. Promising results led to the clinical use of PEA under the brand name Impulsin in former Czechoslovakia [52].

Our work highlights the antiviral activity of PEA by two different mechanisms for the first time. The first one is its binding to SARS-CoV-2 S protein, which our experiments demonstrate to be specific. Indeed, we show that PEA binds SARS-CoV-2 S protein in its RBD in an in vitro cell-based ELISA assay. In support to this evidence, PEA did not prevent infections by VSV or HSV-2, other enveloped viruses that use different glycoproteins and entry receptors (G protein for VSV and gB and gD for HSV-2) to mediate the entry. Surprisingly, we also observed that when PEA was administered above 100 μM, SARS-CoV-2 infection was no longer interfered with. This paradoxical result was not elucidated, but we may hypothesize that this very high concentration, even if it does not lead to cell death, might facilitate in vitro infection through non-canonical viral entry, which is well-described for several enveloped viruses that might alter membrane fusion and/or use alternative receptors [53,54,55]. Second, we show that infected Huh-7 cells treated with PEA dismantle LDs, preventing the usage of these intracellular vesicles by SARS-CoV-2 as a source of energy and as a protection against innate cellular defenses. This is not surprising since PEA activates PPAR-α, a transcription factor that, once activated, generates a cascade of events that lead to the disruption of fatty acid droplets, thereby bringing about LD destruction through β-oxidation. These, in contrast, are essential for the replication of several flaviviruses [38] and also, as recently suggested, for SARS-CoV-2 [25]. As a matter of fact, we observed that the SARS-CoV-2 infection and replication rate increased in Calu-3 cells treated with a potent PPAR-α inhibitor (GW6471). In support of this result, recent evidence demonstrates that SARS-CoV-2 hijacks lipid metabolism in monocytes and other cells, thus accumulating LDs to favor its replication [25].

Our evidence corroborates a major effect of PEA in the inhibition of viral infections. Although the interaction between the S protein and ACE2 is specific for SARS-CoV-2, its activity in LD dismantling and PPAR-α activation might translate into antiviral activity against other RNA viruses that share common replication pathways of SARS-CoV-2, such as ZIKV, WNV, and others. In particular, recent evidence showed that PEA was able to revert the expression of inflammatory markers in murine alveolar macrophages exposed to S protein in a concentration-dependent manner. Moreover, they demonstrated that PPAR-α activation is crucial for the anti-inflammatory activity of PEA. Indeed, macrophages taken from PPAR-α -/- mice did not reduce the pro-inflammatory markers released when treated with PEA at high concentrations [56].

In conclusion, in the present work we demonstrated a novel mechanism of action for PEA as a direct and indirect antiviral agent against SARS-CoV-2. This evidence reinforces the notion that this compound might significantly impact the course of COVID-19. Indeed, considering that the protective effects of PEA in COVID-19 are the current objectives of two clinical trials (NCT04619706 and NCT04568876) and given the relative lack of toxicity of PEA for humans, further preclinical and clinical tests will be needed to fully consider this molecule as a promising adjuvant in the current therapy of COVID-19 or against emerging RNA viruses that share the same route of replication as coronaviruses.

## Figures and Tables

**Figure 1 viruses-14-01080-f001:**
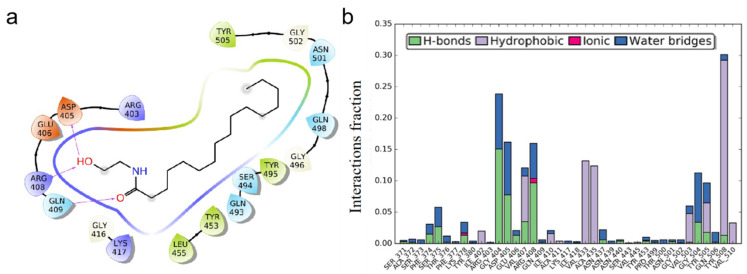
PEA interacts with SARS-CoV-2 S protein. (**a**) Binding mode of 6LZG with PEA after docking: “->” represents hydrogen bond interaction. (**b**) Amino acid residual interaction fractions in interaction with PEA.

**Figure 2 viruses-14-01080-f002:**
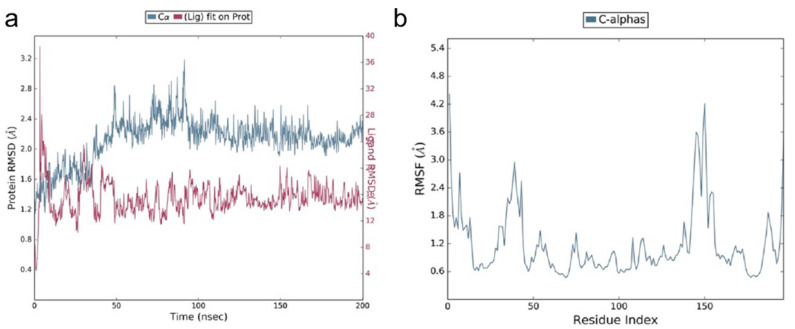
Molecular dynamic simulation study of RBD–PEA complex. (**a**) RMSD profile and (**b**) RMSF profile.

**Figure 3 viruses-14-01080-f003:**
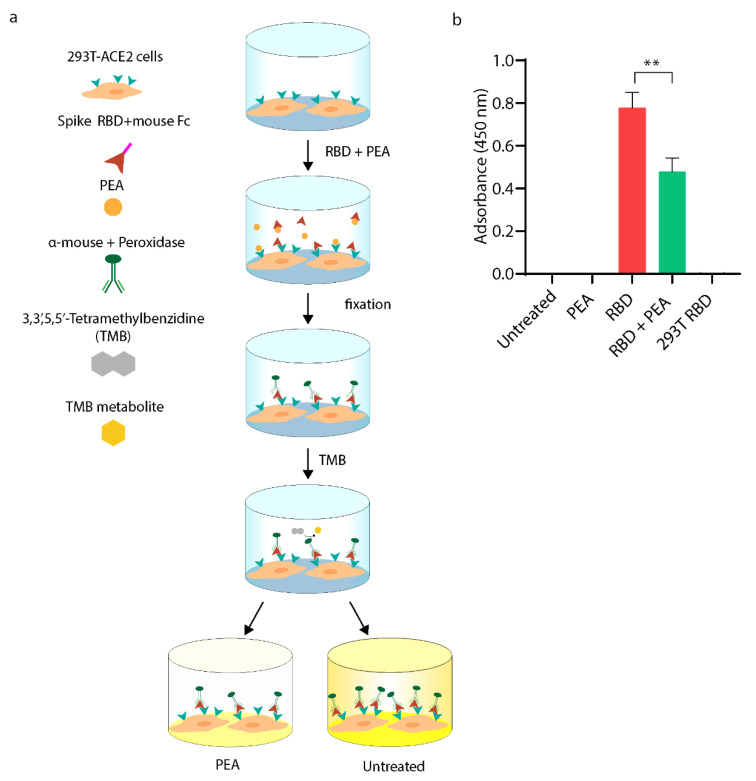
Cell-based ELISA assay performed on 293T-ACE2 cells. (**a**) Schematic illustration of experimental workflow. Briefly, 293T-ACE2 cells were exposed to equimolar concentrations of PEA (10 μM) and recombinant SARS-CoV-2 RBD fused with murine Fc. Then, cells were fixed and stained with peroxidase-labeled α-mouse IgG. (**b**) Statistical analysis of cell-based ELISA assay shown in a. Data are expressed as means ± SD and were analyzed by a one-way ANOVA (N = 4, α = 0.05, ** *p* < 0.001).

**Figure 4 viruses-14-01080-f004:**
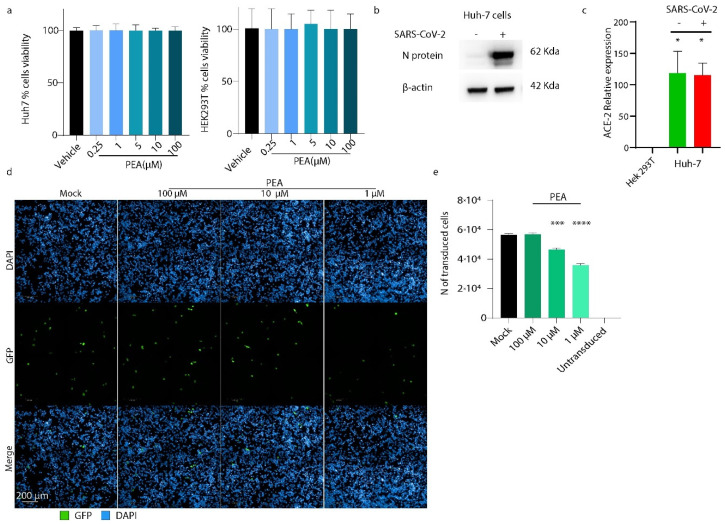
PEA decreases S-pseudotyped lentiviral vector transduction. (**a**) Cell viability assay performed on Huh-7 and 293T cells using WST-8 assay. (**b**) Western blot performed on Huh-7 cells, infected or not with SARS-CoV-2. (**c**) qRT-PCR performed on Hek293T cells and Huh-7 cells. The latter were infected or not with SARS-CoV-2. (**d**) High-content screening of transduced Huh-7 cells expressing GFP protein (green) treated or not with increasing concentrations of PEA. (**e**) Quantification of transduced cells. Around 40 fields were analyzed per well using a 63× water objective. Data are expressed as means ± SD and were analyzed using a one-way ANOVA (n = 3, * *p* < 0.5, *** *p* < 0.001, **** *p* < 0.0001).

**Figure 5 viruses-14-01080-f005:**
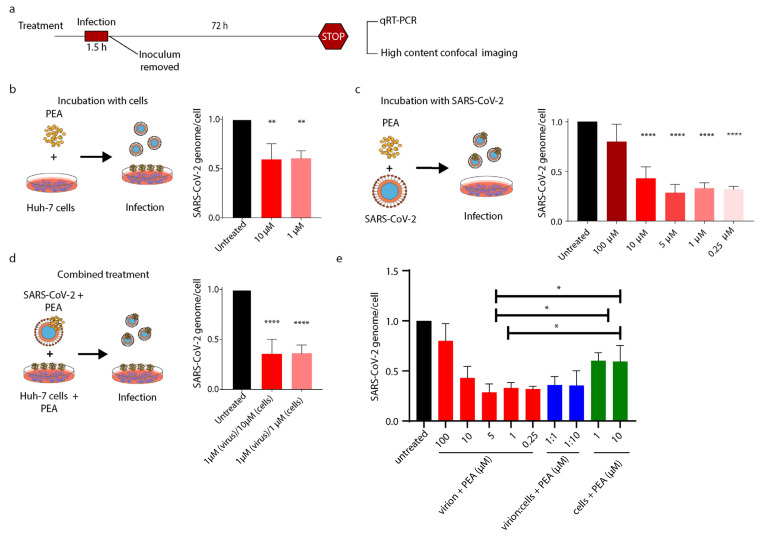
PEA decreases entry of SARS-CoV-2. (**a**) Schematic illustration of experimental workflow and PEA treatment. (**b**) SARS-CoV-2 genome’s relative quantity in cells treated or not with PEA after infection. (**c**) SARS-CoV-2 genome’s relative quantity in cells infected with PEA-preincubated SARS-CoV-2 virions. (**d**) SARS-CoV-2 genome’s relative quantity in cells treated with a combination of previous treatments. (**e**) SARS-CoV-2 genome’s relative quantity in cells treated as described in b, c, and d. Data were normalized on β-actin content in cell lysates. Data are expressed as means ± SD and were analyzed using a one-way ANOVA (n = 3, * *p* < 0.5, ** *p* < 0.01; **** *p* < 0.0001).

**Figure 6 viruses-14-01080-f006:**
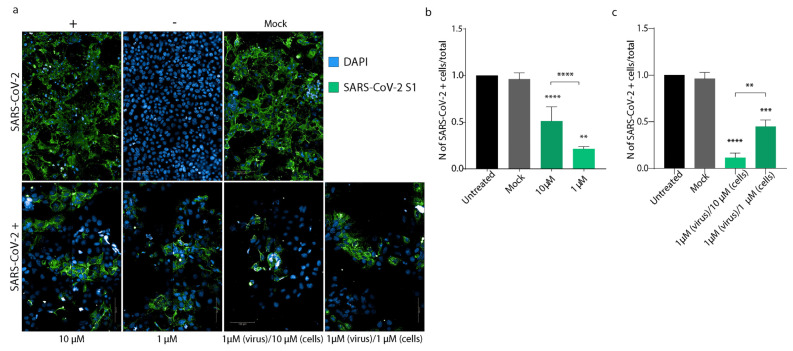
PEA decreases SARS-CoV-2 entry. (**a**) High-content confocal microscopy analysis of SARS-CoV-2-infected cells (green) treated or not with PEA. Representative images of Huh-7 cells infected or not with SARS-CoV-2 and treated or not with PEA 1 and 10 μM. (**b**,**c**) Statistical analyses of the numbers of SARS-CoV-2+ cells, normalized to untreated controls. Ninety fields per well were analyzed using a 40× water objective. Data are expressed as means ± SD and were analyzed by Student’s *t* test (n = 3, ** *p* < 0.01; *** *p* < 0.001; **** *p* < 0.0001).

**Figure 7 viruses-14-01080-f007:**
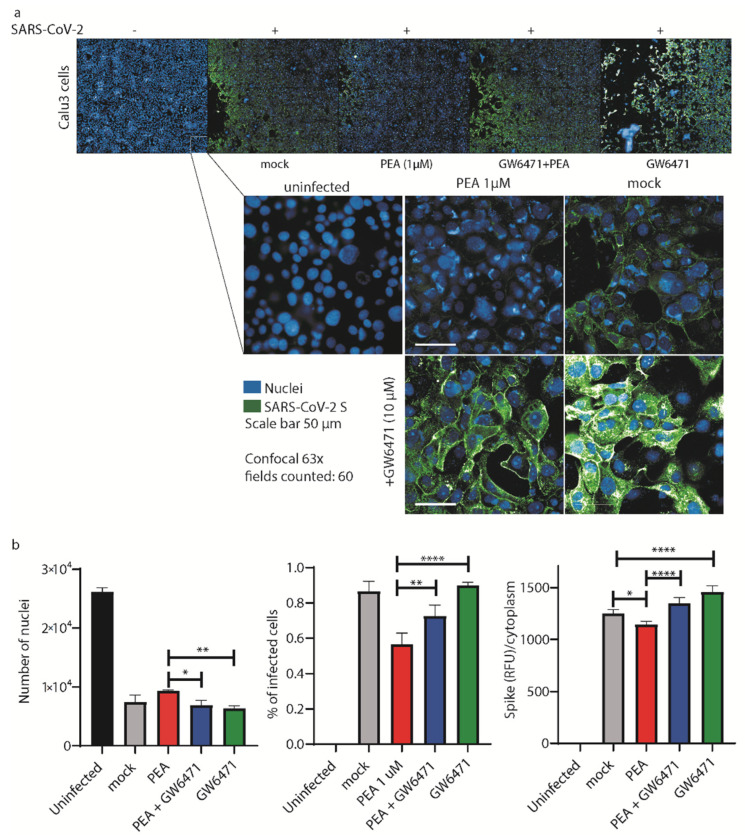
PEA decreases SARS-CoV-2 entry in Calu-3 cells. (**a**) (upper panel) High-content automatic confocal microscopy screening of SARS-CoV-2-infected cells (green) treated or not with PEA and PPAR-α antagonist GW6471 (10 μM). (lower panel) Representative images of Calu-3 cells infected or not with SARS-CoV-2 mixed or not with PEA. Cells were also treated or not with GW6471. Sixty fields were acquired per well. (**b**) Statistical analyses of the number of nuclei detected in the screenings, percentages of infected cells, and the amount of SARS-CoV-2 S protein/cell. Data were analyzed using a 63× water objective. Data are expressed as means ± SD and were analyzed by a one-way ANOVA (n = 3, * *p* < 0.5, ** *p* < 0.01, **** *p* < 0.0001).

**Figure 8 viruses-14-01080-f008:**
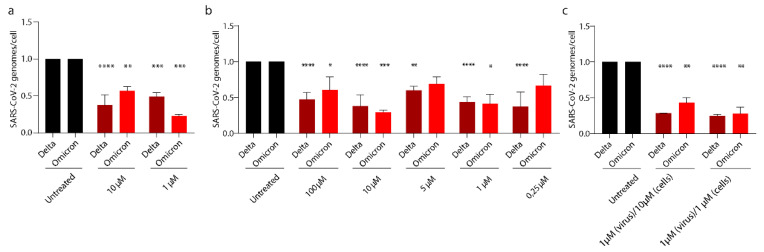
PEA exerts antiviral activity on SARS-CoV-2 VOCs. SARS-CoV-2 relative quantity in (**a**) cells infected with PEA-incubated SARS-CoV-2, (**b**) cells treated or not with PEA, (**c**) both SARS-CoV-2 and cells treated with PEA. Intracellular viral RNA was normalized using β-actin as a housekeeping gene. Data are expressed as means ± SD and were analyzed using a one-way ANOVA (n = 3, * *p* < 0.5, ** *p* < 0.01; *** *p* < 0.001, **** *p* < 0.0001).

**Figure 9 viruses-14-01080-f009:**
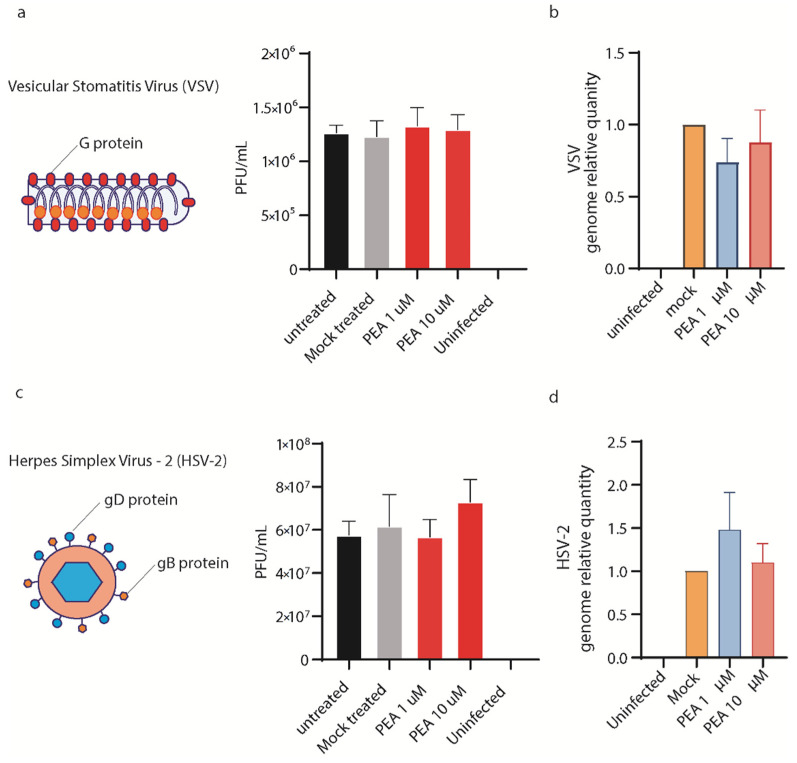
PEA does not reduce VSV and HSV-2 entry in Huh-7 cells. (**a**) (left panel) Schematic illustration of VSV. (right panel) Virus yield reduction assay performed using PEA at 1 and 10 μM. (**b**) VSV genome relative quantity measured by qRT-PCR in cells treated or not with PEA after infection. (**c**) (left panel) Schematic illustration of HSV-2. (right panel) Virus yield reduction assay performed using PEA at 1 and 10 μM. (**d**). HSV-2 genome relative quantity measured by qRT-PCR in cells treated or not with PEA after infection. Data are expressed as means ± SD and were analyzed using a one-way ANOVA (n = 3).

**Figure 10 viruses-14-01080-f010:**
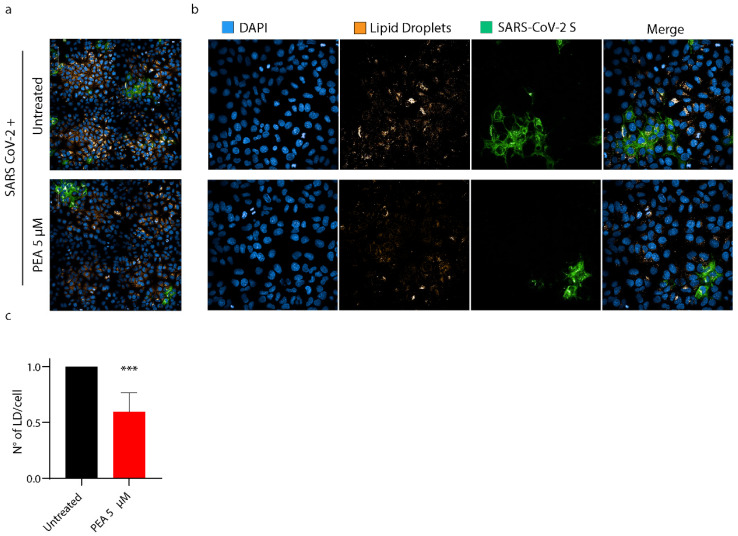
PEA decreases the amount of intracellular LDs. (**a**,**b**) High-content confocal microscopy imaging of SARS-CoV-2-infected Huh-7 cells treated or not with PEA (5 μM). SARS-CoV-2-infected cells are marked in green, LDs were stained with Oil Red O (orange), and nuclei are marked in blue (DAPI). (**c**) Statistical analysis of the numbers of LDs, normalized to the numbers of cells. Sixty fields per well were analyzed using a 40× water objective. Data are expressed as means ± SD and were analyzed by Student’s *t* test (n = 6, *** *p* < 0.001).

## Data Availability

High-content confocal screenings are available upon request to the corresponding author.

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
