# Peer review of "Palmitoylethanolamide (PEA) Inhibits SARS-CoV-2 Entry by Interacting with S Protein and ACE-2 Receptor"

_viruses, 2022, doi:10.3390/v14051080_

Round 1

Reviewer 1 Report

The current state of the manuscript has major shortcomings that need to be addressed.

1) Additional experiments are required to determine the function of PEA in the inhibition of SARS-CoV2. The entry of the virus was suppressed in the treatment of PEA. The data should be addressed which site is a preferred target of PEA to inhibit entry of virus among the ACE2 receptor or Spike protein.

2) All of the data showed that only huh7 cells were used to show infectivity of SARS-CoV-2. The Vero, caco2, or Calu3 cells are more acceptable for observing virus replication of SARS-CoV2. The reasons should be explained why used huh7 cells to observe viral infection in this experiment, and determine viral and cellular protein expression such as ACE2, TMPRSS2, or nucleocapsid protein of SARS-CoV-2 in huh7 cells.

3) For Fig. 8, the additional data are needed to support results. The results should be verified by using PPAR-a KO cells or treatment of anti-PPAR-a to explain specific results against PEA in SARS-CoV-2 infection.

4) All references for the statement should be provided with original studies.: Line 51,80, 83, 85

5) In fig. 6, the authors should describe what is the differences of recognized sites against PEA in Delta or Omicron. Delta and Omicron has mutations on Spike protein. You should explain the possible effect against affinity of spike-PEA complex such as stability of hydrogen bond in complex by infection of SARS-CoV-2 variants.

6) In fig 6(b), the authors should explain how to determine released virion in the virus-infected cells.

7) For Fig. 4, the authors should describe whether the virus or PEA was washed out or not after combined treatment.

8) The additional data are needed to determine the cytotoxicity of PEA according to the treatment of 1, 10, and 100uM in huh7 and 293T cells.

9) Line 190: Please add catalog number for antibody.

10) Fig 1: Please, explain what is represented by color in the graph (B).

11) Line 270: Please check the correct figure number (Figure5 -> Figure 3).

12) All of the results did not show antiviral effects in a dose-dependent manner of PEA. Additional experiments are required to rearrange of concentration of PEA to observe antiviral efficacy in a dose-dependent manner or, should explain why antiviral efficacy was not observed in high concentrations (10 or 100uM).

13) Line 346 and 360: Please check the correct number (3.4 ->3.5)

14) In Fig 4, there are no data to demonstrate the antiviral efficacy of combined treatment compared to non-combined treatment. These results should be represented in one graph to compare each antiviral efficacy against control.

15) For figure 5(a), this study should be provided another or whole well image to identify viral infection (0.1 MOI and 3dpi). SARS-CoV-2 (+) image is represented that 100% of cells were infected with SARS-CoV-2 (expressed green signal).

Author Response

The current state of the manuscript has major shortcomings that need to be addressed.

  • Additional experiments are required to determine the function of PEA in the inhibition of SARS-CoV2. The entry of the virus was suppressed in the treatment of PEA. The data should be addressed which site is a preferred target of PEA to inhibit entry of virus among the ACE2 receptor or Spike protein.

We appreciate this observation. We have added a new result that shows cell-based ELISA assays performed on 293T-ACE2 cells exposed to SARS-CoV-2 S RBD protein together or not with PEA (figure 3).Line 281-288 now states ” With the aim to confirm the predicted binding of PEA with SARS-CoV-2 S protein, local-ized in the RBD region, we took advantage of a cell-based ELISA assay, schematically il-lustrated in figure 3a. Briefly, we exposed 293T-ACE2 cells to SARS-CoV-2 recombinant RBD protein, fused with murine Fc, with or without equimolar concentration of PEA. Then, cells were washed, fixed and stained with labeled α-mouse antibody. As shown in figure 3b, PEA decreases RBD binding with ACE2 by ~50% if compared to mock-treated counterparts. As expected,  no signal was detected on 293T WT cells.”

2) All of the data showed that only huh7 cells were used to show infectivity of SARS-CoV-2. The Vero, caco2, or Calu3 cells are more acceptable for observing virus replication of SARS-CoV2. The reasons should be explained why used huh7 cells to observe viral infection in this experiment, and determine viral and cellular protein expression such as ACE2, TMPRSS2, or nucleocapsid protein of SARS-CoV-2 in huh7 cells.

Done. We added two experiments confirming Huh-7 as good model for SARS-CoV-2 infection. We also added references supporting this point. Moreover, we repeated high-content confocal quantification on Calu3 cells, as reviewer asked. Lines 370-387

3) For Fig. 8, the additional data are needed to support results. The results should be verified by using PPAR-a KO cells or treatment of anti-PPAR-a to explain specific results against PEA in SARS-CoV-2 infection.

Done. PPAR-α increases SARS-CoV-2 replication. We used the antagonist GW6471, experiments are now included in figure 7

4) All references for the statement should be provided with original studies.: Line 51,80, 83, 85

Done.

5) In fig. 6, the authors should describe what is the differences of recognized sites against PEA in Delta or Omicron. Delta and Omicron has mutations on Spike protein. You should explain the possible effect against affinity of spike-PEA complex such as stability of hydrogen bond in complex by infection of SARS-CoV-2 variants.

Done. Lines 245-248 now states “Earlier studies from Suresh Kumar et al 2021, there is no RBD mutations found in 405, 408 and 409 residues indicating efficacy of PEA binding to the RBD domain remains unchanged with delta (2 mutations) and omicron (15 mutations) variants”

6) In fig 6(b), the authors should explain how to determine released virion in the virus-infected cells.

We apologize for the misleading statement. Line 363 now states” virions reduced the number of viral genomes/cell up to nearly 65% for both VOCs with…”

7) For Fig. 4, the authors should describe whether the virus or PEA was washed out or not after combined treatment.

We apologize for the lack of information. Viruses in combination or not with PEA were removed. We added line 140 in materials and methods “inocula were removed by three washings with PBS”

8) The additional data are needed to determine the cytotoxicity of PEA according to the treatment of 1, 10, and 100uM in huh7 and 293T cells.

Done. Figure 4a shows toxicity assays performed with WST-8 on Huh-7 and 293T cells

9) Line 190: Please add catalog number for antibody.

Done. Catalog number #40588 RC02 is now added

10) Fig 1: Please, explain what is represented by color in the graph (B).

Done. please see revised Figure 1.

11) Line 270: Please check the correct figure number (Figure5 -> Figure 3).

Done.

12) All of the results did not show antiviral effects in a dose-dependent manner of PEA. Additional experiments are required to rearrange of concentration of PEA to observe antiviral efficacy in a dose-dependent manner or, should explain why antiviral efficacy was not observed in high concentrations (10 or 100uM).

Done. This task was analyzed by first assessing that PEA administered at 100 μM does not exert any toxicity (Figure 4a). While this concentration is extremely high and far from clinical application, we suppose that this concentration might facilitate in vitro infection trough non canonical viral entry. Lines 463-467 were added “Surprisingly, we also observed that when PEA was administered above 100 μM, SARS-CoV-2 infection was no longer interfered with. This paradoxical result was not elucidated but we may hypothesize that this very high concentration, even if it does not lead to cell death, might facilitate in vitro infection through non-canonical viral entry.

13) Line 346 and 360: Please check the correct number (3.4 ->3.5)

Done.

14) In Fig 4, there are no data to demonstrate the antiviral efficacy of combined treatment compared to non-combined treatment. These results should be represented in one graph to compare each antiviral efficacy against control.

Done.

15) For figure 5(a), this study should be provided another or whole well image to identify viral infection (0.1 MOI and 3dpi). SARS-CoV-2 (+) image is represented that 100% of cells were infected with SARS-CoV-2 (expressed green signal).

A new set of experiments is now present, performed on Calu-3 cells as reviewer asked. This acquisition was now analyzed also providing the absolute number of infected cells per treatment (figure 7).

Reviewer 2 Report

In this manuscript the authors show that PEA inhibits SARS-CoV-2 entry by 
interacting with S protein and ACE-2 receptor. The interactions are analysed in silico. To demonstrate their findings, the authors mix PEA with the virus (for interaction with S) or pretreat cells (for interaction with ACE-2) with PEA.  The manuscript is well written, however, I doubt if the results prove the claims made.

  • The interaction of PEA with S-Protein and ACE has only been demonstrated in silico. To demonstrate such interactions, other methods such as ELISA and competition assays with the protein have to be performed. Mixing virus with PEA and having then a reduced infectivity dosn't prove binding of PEA to S. Similarily, pretreating cells with PEA and having then a reduced infectivity doesn's show that PEA binds to the main receptor of SARS-CoV-2 (ACE-2). It might as well inhibit other steps of viral uptake or block a co-receptor. 
  • The pseudovirion assays in figure 3 are not convincing. The fact that the highest concentration of the drug has no effect on infectivity at all has to be discussed.
  • The authors use for SARS-CoV-2 only realtime PCR and no plaque/focus assays, whereas for the control viruses (HSV and VSV) they use plaque assay. They should use the same assays for all viruses. 
  • Reference 29 and 30 are the same. 

Author Response

In this manuscript the authors show that PEA inhibits SARS-CoV-2 entry by 
interacting with S protein and ACE-2 receptor. The interactions are analysed in silico. To demonstrate their findings, the authors mix PEA with the virus (for interaction with S) or pretreat cells (for interaction with ACE-2) with PEA.  The manuscript is well written, however, I doubt if the results prove the claims made.

  • The interaction of PEA with S-Protein and ACE has only been demonstrated in silico. To demonstrate such interactions, other methods such as ELISA and competition assays with the protein have to be performed. Mixing virus with PEA and having then a reduced infectivity dosn't prove binding of PEA to S. Similarily, pretreating cells with PEA and having then a reduced infectivity doesn's show that PEA binds to the main receptor of SARS-CoV-2 (ACE-2). It might as well inhibit other steps of viral uptake or block a co-receptor.  

We appreciate this observation. We have added a new result that shows cell-based ELISA assays performed on 293T-ACE2 cells exposed to SARS-CoV-2 S RBD protein together or not with PEA (figure 3).Line 281-288 now states ” With the aim to confirm the predicted binding of PEA with SARS-CoV-2 S protein, local-ized in the RBD region, we took advantage of a cell-based ELISA assay, schematically il-lustrated in figure 3a. Briefly, we exposed 293T-ACE2 cells to SARS-CoV-2 recombinant RBD protein, fused with murine Fc, with or without equimolar concentration of PEA. Then, cells were washed, fixed and stained with labeled α-mouse antibody. As shown in figure 3b, PEA decreases RBD binding with ACE2 by ~50% if compared to mock-treated counterparts. As expected,  no signal was detected on 293T WT cells.”

  • The pseudovirion assays in figure 3 are not convincing. The fact that the highest concentration of the drug has no effect on infectivity at all has to be discussed.

Done. This task was analyzed by first assessing that PEA administered at 100 μM does not exert any toxicity (Figure 4a). While this concentration is extremely high and far from clinical application, we suppose that this concentration might facilitate in vitro infection trough non canonical viral entry. Lines 463-467 were added “Surprisingly, we also observed that when PEA was administered above 100 μM, SARS-CoV-2 infection was no longer interfered with. This paradoxical result was not elucidated but we may hypothesize that this very high concentration, even if it does not lead to cell death, might facilitate in vitro infection through non-canonical viral entry.

  • The authors use for SARS-CoV-2 only realtime PCR and no plaque/focus assays, whereas for the control viruses (HSV and VSV) they use plaque assay. They should use the same assays for all viruses. 

Done. We added released genome quantification by qRT-PCR for both VSV and HSV-2 as for SARS-CoV-2. Line 351-353 now states “As shown in Figure 7a-d PEA did not affect viral entry for either virus, measured as plaque forming units/mL and by qRT-PCR quantification of viral genomes in the supernatants.”

Round 2

Reviewer 1 Report

1) How are RNA levels in SARS-CoV-2 genomes/ cells calculated from qPCR data? What do they mean? Viral RNA levels should be at some way put into context of the cellular RNA levels. 

2) Add references to support your hypothesis in line 466.

3) Add primers information for ACE2  in material and method.

4) Add Cell-based ELISA and Cell viability assay section in material and method.

Author Response

  • How are RNA levels in SARS-CoV-2 genomes/ cells calculated from qPCR data? What do they mean? Viral RNA levels should be at some way put into context of the cellular RNA levels. 

Sorry for the lack of clarity. Β-actin primers were used to normalize the viral genomes in cell lysates. Figure 5 and Figure 8 now states “Data were normalized on β-actin content in cell lysates”.

  • Add references to support your hypothesis in line 466.

Done.

3) Add primers information for ACE2  in material and method.

done

4) Add Cell-based ELISA and Cell viability assay section in material and method.

Done see lines 136-140 and 214-226

Reviewer 2 Report

All concerns have been adressed.

Author Response

We thank the reviewer for his help. The manuscript was reviewed by a native english speaker